# Similar Outcomes and Satisfaction of the Proprioceptive versus Standard Training on the Knee Function and Proprioception, Following the Anterior Cruciate Ligament Reconstruction

Paweł Bąkowski [1] , Kinga Ciemniewska-Gorzela [1], Kamilla Bąkowska-Żywicka [2], Łukasz Stołowski [1] and Tomasz Piontek [1,3,*]

1   Department of Orthopedic Surgery, Rehasport Clinic, 60201 Poznan, Poland; pawel.bakowski@rehasport.pl (P.B.); kinga.ciemniewska-gorzela@rehasport.pl (K.C.-G.); lukasz.stolowski@rehasport.pl (Ł.S.)
2   Institute of Bioorganic Chemistry, Polish Academy of Sciences, 61704 Poznan, Poland; kamilla.bakowska-zywicka@rehasport.pl
3   Department of Spine Disorders and Pediatric Orthopedics, University of Medical Sciences Poznan, 61545 Poznan, Poland
*   Correspondence: tomasz.piontek@rehasport.pl

**Abstract:** Background: Patients with anterior cruciate ligament (ACL) injuries have poorer proprioception than people without such injuries. The effects of proprioceptive training on knee functionality and proprioceptive improvement after ACL reconstruction is still unclear. Methods: The research material consisted of 40 patients after ACL reconstruction. Of the 40 patients, 20 of them were rehabilitated with a standard program and 20 with additional proprioceptive exercises. The subjective and the objective scores were evaluated. Results: No differences were found between the proprioceptive versus the conservative postoperative rehabilitation in the functional nor in the proprioception outcomes of the operated limbs. Conclusions: There is no advantage to function in doing proprioceptive rehabilitation exercises following the ACL reconstruction, when compared with a traditional strengthening program.

**Keywords:** ACL reconstruction; rehabilitation; proprioception

## 1. Introduction

It has long been thought that anterior cruciate ligament (ACL) injuries can be detrimental to the proprioception of the knee [1]. A recent meta-analysis by Relph et al., involving 191 ACL-injured patients and 82 controls, revealed that the patients with ACL injury had poorer proprioception than people without such injuries [2]. Furthermore, the proprioceptive changes in the contralateral knee joint following ACL injury have also been reported [3].

It is also well-considered that the time taken and the safety of returning to the normal daily activities and to sport after the ACL reconstruction depends on the rehabilitation protocol [4–6]. Evidence suggests that an accelerated rehab protocol without postoperative bracing, in which reduction in pain, swelling and inflammation and regaining range of motion, strength and neuromuscular control are the most important aims, has important advantages and does not lead to stability problems [7]. However, considering the large differences in the clinical and outpatient protocols, there is no consensus on such rehabilitation programs.

Nowadays, proprioceptive training has been demonstrated in the world and has been widely used for patients with an ACL reconstruction [8,9]. Recent trends in ACL rehab are consistently using proprioceptive training with mixed results [10]. Therefore, further work is required to provide evidence. Considering all of the above, the purpose of this

work was to determine the effectiveness of proprioceptive versus standard postoperative rehabilitation on the knee function and proprioception, following the ACL reconstruction.

## 2. Materials and Methods

### 2.1. Patients

Patients >6 months post injury who underwent ACL reconstruction using a hamstring graft (semitendinosus and gracilis muscles) at the Department of Pediatric Orthopedics, Medical University in Poznań by one experienced orthopedic surgeon from the year 2005 to 2007 were included in the study. Patients with coexisting damage to another ligament of the knee joint, a history of trauma in another joint or trauma to the other limb were excluded.

Patients were randomized to two groups—group A, which consisted of 20 patients rehabilitated with a protocol with proprioceptive elements, and group B, which consisted of 20 patients rehabilitated without proprioceptive elements during training.

Randomization was done by drawing.

### 2.2. Surgical Protocol and Postoperative Management

The surgical technique was previously described in detail, in Polish, by Piontek et al. [11]. Shortly, the surgery was performed under spinal anesthesia. Patients were placed on their backs with a tourniquet on the operated thigh. Two standard arthroscopic techniques were used: Endo-button and Rigid-fix. In all patients, a four-strand hamstring tendon autograft (semitendinosus and/or gracilis tendon) was used. Graft fixation was obtained by using the loop system and the plate on the cortex of the femur (Endo-button) or bioscrews (Rigid-fix). During the procedure, the ACL stumps were removed.

The operated limb was placed in an orthosis, stabilizing the knee in a 0–30° flexion. The postoperative hospitalization lasted no longer than two days. The management of the pain and swelling in the initial hours after the surgery was executed by the means of cold packs, limb elevation and administration of analgesics. Starting from the second day, the post-surgery patients could walk with the elbow crutches and without weight-bearing on the operated limb. The knee remained immobilized in the postoperative brace for a period of 2–3.5 weeks.

### 2.3. Rehabilitation Program

The rehabilitation program for both groups was divided into five stages, moving forward between them according to time and patient's function. A detailed description of both rehabilitation programs is presented in Table 1.

In the first stage, isometric exercises of the quadriceps, hamstrings, adductors and abductors were introduced on post-op day 2. The exercises increasing the range of motion were introduced a day after the surgery and lasted till the 6th week.

In the second stage, from the 3rd to the 8th week, patients started to increase weight bearing and after 6 weeks were recommended not to use the elbow crutches. Additionally, active exercises in the lying position and hip exercises in the one-leg standing position were introduced. Core stability exercises, e.g., plank and side plank, were introduced. The proprioceptive exercises were introduced in group A.

In the third stage, standing exercises were introduced. In group A, patients learned to properly load the operated limb on the dynamometric platform, which showed ground reaction force. Additionally, standing on unstable ground and squats up to 90 degrees on stable and unstable ground were performed.

In the fourth stage, from the 13th till the 24th week, an external load was added to the previously performed exercises, depending on the condition and the patient's abilities. The warm-up was continued on a stationary bike with an increasing load and the exercises were in closed chains. The patients from group A performed the exercises on stable and unstable ground (e.g., one-leg squat, lunges, reverse lunges and one-leg deadlifts).

The fifth stage, from the 24th week, included double-leg jumps, single-leg jumps and plyometric exercises (jumping exercises aimed at increasing dynamics and power).

**Table 1.** Summary of the rehabilitation programs.

| Stage (Duration) | Group A—Proprioceptive Exercises | Group B—Standard Protocol |
|---|---|---|
| I (0–2 weeks) | i. isometric exercises of the quadriceps, hamstrings, adductors and abductors ii. exercises increasing the range of motion | i. isometric exercises of the quadriceps, hamstrings, adductors and abductors ii. exercises increasing the range of motion |
| II (3–8 weeks) | i. active exercises in the lying position ii. hip exercises in the one-leg standing positioncore stability exercises, e.g., plank and side plank iii. forcing the ball into the wall of the operated limb while lying on the back iv. extending the knee joint with an external load | i. active exercises in the lying position ii. hip exercises in the one-leg standing positioncore stability exercises, e.g., plank and side plank |
| III (9–12 weeks) | i. standing exercises ii. dynamometric platforms iii. standing on unstable ground iv. squats up to 90 degrees on unstable ground | i. standing exercises |
| IV (13–24 weeks) | i. external load ii. warm-up on a stationary bike with an increasing load iii. exercises in closed chains iv. exercises on stable and unstable ground (e.g., one-leg squat, lunges, reverse lunges and one-leg deadlifts) v. both-leg and one-leg jumping on a trampoline with stops | i. external load ii. warm-up on a stationary bike with an increasing load iii. exercises in closed chains |
| V (from the 24th week) | i. double-leg jumps, single-leg jumps and plyometric exercises (jumping exercises aimed at increasing dynamics and power) | i. double-leg jumps, single-leg jumps and plyometric exercises (jumping exercises aimed at increasing dynamics and power) |

*2.4. Evaluations*

The functional and the clinical outcomes of the two groups were compared. All patients were assessed postoperatively. The subjective outcomes were evaluated using an International Knee Documentation Committee knee ligament healing standard form (IKDC 2000) and the Lysholm knee scoring scale. The mechanical knee stability was assessed with the Lachman test and the drawer test. Visual-proprioceptive control was performed using the dynamic and static Riva tests during standing on one leg (single stance, as in [12]) on a Delos Postural Proprioceptive System. Isokinetic evaluation of extensor and flexor muscles was performed using a Biodex 3 dynamometer.

*2.5. Statistical Analysis*

The statistical analysis was performed using Statistica v. 7.1 software. The normality of the distribution was verified using the Shapiro–Wilk test. There was no normal distribution and hence the quantitative variables were presented using the median $\pm$ standard deviation. The Wilcoxon signed rank test was used to determine the significant differences between objective and subjective parameters. The correlation between results was determined using the Spearman's rank correlation test. Statistical significance was set at $p < 0.05$.

## 3. Results

### 3.1. Demographics

Forty patients were included in the study, 33 of which were males and 7 were females. The mean age was 32.4 (range 14–57). The mean BMI was 22.69 (range 18.23–28.67).

### 3.2. Subjective Knee Evaluation

The distribution of the IKDC subjective knee scores are presented in Table 2. The mean value of the IKDC form for group B was lower than "good" (<76). The mean value for group A was higher than for group B in the IKDC and Lysholm test (not significant). A strong correlation was noticed between the Lysholm and IKDC scores (r = 0.827, $p$ = 0.000 for group A and r = 0.81, $p$ = 0.000 for group B).

**Table 2.** Mechanical knee stability in anterior cruciate ligament (ACL) patients. Values are presented as median ± standard deviation. The minimum and maximum values are given in brackets. $p$ values are indicated. IKDC—International Knee Documentation Committee score.

| Test | Group A | Group B | $p$ |
|---|---|---|---|
| IKDC | 76.2 ± 10.2 (54.0–87.0) | 75.1 ± 7.8 (59.0–86.0) | 0.398 |
| Lysholm | 93.5 ± 10.2 (66.0–100.0) | 90.3 ± 9.7 (65.0–100.0) | 0.091 |

We found that the self-assessment of the knee joint function depended on the body mass index (BMI) and on the age of the patients. A strong correlation was noticed between the BMI and the Lysholm scale (r = −0.332, $p$ = 0.036 in group A and r = −0.512, $p$ = 0.021 for group B) as well as between the BMI and IKDC (r = -0.388, $p$ = 0.013 in group A and r = −0.6508, $p$ = 0.022 for group B). A prominent correlation was also noticed between the age and the Lysholm scale (r = −0.354, $p$ = 0.025 in group A and r = −0.677, $p$ = 0.001 for group B) as well as between the age and IKDC (r = −0.372, $p$ = 0.018 in group A and r = −0.609, $p$ = 0.004 for group B).

### 3.3. Objective Knee Evaluation

After the ACL rehabilitation program with the proprioceptive elements, patients (group A) had significantly higher values in the relative isokinetic knee flexors ($p$ = 0.024) and extensors ($p$ = 0.044) in the intact limb, in comparison to the patients rehabilitated without proprioception elements (group B, Table 3). However, there were no statistically important differences between the groups regarding the absolute isokinetic strength and the knee flexor/extensor ratio. Higher values for the Lachman and the drawer tests were observed for the patients rehabilitated with the proprioception exercises (Table 4). However, they were not significantly important.

In a comparative analysis of the visual proprioceptive control results, the distribution of the parameters for groups A and B did not differ significantly (Table 5). Very good (4.1–5.0) and excellent (5.1–6.0) results were observed in the static Riva tests (STR). In the dynamic test (DTR, the results were very good. However, the cases of insufficient level occurred more frequently in group B.

A strong negative correlation was observed between the visual proprioceptive control results and the isokinetic parameters for the operated limb: peak torque/weight for extensors muscles (Ext) (r = −0.609, $p$ = 0.00) and flexors muscles (Flx)/Ext ratio (r = −0.681, $p$ = 0.001).

**Table 3.** Absolute and relative isokinetic strength in ACL patients. Values are presented as median $\pm$ standard deviation. The minimum and maximum values are given in brackets. *p* values are indicated. PkTrq—peak torque; Ext—extensors muscles; Flx—flexors muscles; Pw—peak torque/body weight ratio.

| Test | Limb | Group A | Group B | *p* |
|---|---|---|---|---|
| 60-PkTrq Ext | Operated | 171 $\pm$ 52 (70–249) | 162 $\pm$ 50 (86–256) | 0.512 |
| | Intact | 209 $\pm$ 54 (113–300) | 188 $\pm$ 43 (110–247) | 0.296 |
| 60-PkTrq Flx | Operated | 100 $\pm$ 31 (47–153) | 99 $\pm$ 31 (41–153) | 0.678 |
| | Intact | 114 $\pm$ 25 (62 –153) | 104 $\pm$ 28 (68–151) | 0.175 |
| PkTrq/weight Ext | Operated | 230 $\pm$ 66 (121–357) | 202 $\pm$ 74 (69–330) | 0.277 |
| | Intact | 280 $\pm$ 62 (199–411) | 228 $\pm$ 57 (75–309) | 0.044 |
| PkTrq/weight Flx | Operated | 134 $\pm$ 38 (59–219) | 122 $\pm$ 40 (39–192) | 0.445 |
| | Intact | 154 $\pm$ 31 (101–218) | 126 $\pm$ 37 (40–189) | 0.024 |
| Mean Pw Ext | Operated | 117 $\pm$ 34 (49–182) | 108 $\pm$ 31 (52–162) | 0.414 |
| | Intact | 137 $\pm$ 36 (74–197) | 127 $\pm$ 30 (76–174) | 0.550 |
| Mean Pw Flx | Operated | 68 $\pm$ 22 (27–108) | 68 $\pm$ 24 (28–112) | 0.904 |
| | Intact | 78 $\pm$ 19 (45–114) | 78 $\pm$ 19 (45–104) | 0.194 |
| Flx/Ext Ratio [%] | Operated | 59 $\pm$ 11 (46–80) | 63 $\pm$ 18 (40–121) | 0.602 |
| | Intact | 56 $\pm$ 8 (44–76) | 55 $\pm$ 10 (41–78) | 0.989 |

**Table 4.** Mechanical knee stability in ACL patients. Values are presented as mean $\pm$ standard deviation. The minimum and maximum values are given in brackets. *p* values are indicated.

| Test | Limb | Group A | Group B | *p* |
|---|---|---|---|---|
| Lachmann | Operated | 7.4 $\pm$ 1.8 (4.0–10.0) | 7.0 $\pm$ 1.7 (4.0–11.0) | 0.445 |
| | Intact | 6.5 $\pm$ 2.3 (4.0–12.0) | 6.2 $\pm$ 1.4 (4.0–10.0) | 0.947 |
| Drawer test | Operated | 7.0 $\pm$ 1.5 (4.0–10.0) | 6.8 $\pm$ 1.7 (3.0–11.0) | 0.678 |
| | Intact | 6.1 $\pm$ 1.5 (4.0–8.0) | 6.3 $\pm$ 1.7 (3.0–10.0) | 0.758 |

**Table 5.** The visual proprioceptive control results. Values are presented as median $\pm$ standard deviation. The minimum and maximum values are given in brackets. *p* values are indicated. STRD_EO—deviations from the resultant of static Riva test (STR) mean axes, eyes open; STRD_EC—deviations from the resultant of STR mean axes, eyes closed; RF—risk fall; DTRD—deviations from the resultant of dynamic Riva test (DTR) mean axes; IS—instability of the human-platform system; and VPC—visual-proprioceptive control coefficient.

| Test | Limb | Group A | Group B | *p* |
|---|---|---|---|---|
| STRD_EO | Operated | 5.5 $\pm$ 0.6 (4.0–6.0) | 5.0 $\pm$ 1.7 (4.0–6.0) | 0.478 |
| | Intact | 5.7 $\pm$ 0.5 (4.0–6.0) | 5.5 $\pm$ 1.2 (1.0–6.0) | 0.728 |
| STRD_EC | Operated | 4.3 $\pm$ 1.2 (1.0–6.0) | 3.9 $\pm$ 1.6 (1.0–5.5) | 0.698 |
| | Intact | 4.0 $\pm$ 1.5 (1.0–6.0) | 4.1 $\pm$ 1.8 (1.0–6.0) | 0.647 |
| DTRD | Operated | 4.5 $\pm$ 0.5 (3.5–5.5) | 4.4 $\pm$ 0.7 (3.0–6.0) | 0.779 |
| | Intact | 4.5 $\pm$ 0.4 (4.0–5.0) | 4.4 $\pm$ 0.7 (3.0–5.5) | 0.607 |
| RF | Operated | 1.6 $\pm$ 0.8 (0.0–2.0) | 1.5 $\pm$ 0.8 (0.0–2.0) | 0.495 |
| | Intact | 1.5 $\pm$ 0.8 (0.0–2.0) | 1.4 $\pm$ 0.8 (0.0–2.0) | 0.380 |
| IS | Operated | 4.4 $\pm$ 1.8 (1.5–9.4) | 5.3 $\pm$ 2.2 (1.4–11.3) | 0.134 |
| | Intact | 4.6 $\pm$ 1.9 (2.2–9.2) | 5.2 $\pm$ 2.7 (2.0–10.2) | 0.627 |
| VPC | Operated | 54.4 $\pm$ 12.9 (32.3–73.8) | 51.6 $\pm$ 12.5 (31.7–76.9) | 0.529 |
| | Intact | 53.9 $\pm$ 14.2 (18.0–73.4) | 53.8 $\pm$ 14.5 (27.4 –74.3) | 0.945 |

### 3.4. Correlations between Subjective and Objective Tests

The Spearman Rho coefficient was used to test the correlation between the scales. The correlations between subjective and objective tests were analyzed for the whole study group (n = 40). The results of the subjective score assessments for the operated limb correlated well with the results of the isokinetic strength parameters as well as the visual proprioceptive control results (Table 6).

**Table 6.** Subjective-objective tests correlation. r—Spearman Rho coefficient. *p* values are indicated. STRD_EC—deviations from the resultant of STR mean axes, eyes closed; VPC—Visual-proprioceptive control coefficient.

| Subjective Test | Objective Test | r | *p* |
|---|---|---|---|
| IKDC | Pk Trq/weight Ext | 0.615 | 0.004 |
| IKDC | Pk Trq—weight Ext | −0.614 | 0.004 |
| IKDC | VPC | −0.667 | 0.001 |
| Lachman | STRD_EC | −0.645 | 0.002 |

## 4. Discussion

The main finding of this study was that no differences were found between the proprioceptive versus the conservative postoperative rehabilitation, in the functional nor in the proprioception outcomes of the operated limbs. Although the rehabilitation program with the proprioceptive elements provided better knee stability, the difference was clinically meaningless. Therefore, the results of the current study suggest that there is no advantage to function in doing proprioceptive rehabilitation exercises following the ACL reconstruction, when compared with a traditional strengthening program.

The results of the current study are similar to those of Liu-Ambrose et al. [13] and Cooper et al. [14]. Liu-Ambrose et al. investigated patients after semitendinosus tendon ACL reconstruction. 10 participants were enrolled in a 12-week training program: five were assigned to the proprioceptive program and five to the isotonic one. Cooper et al. investigated 19 patients during 14 weeks: 14 were rehabilitated with the proprioception exercises and 15 with the strengthening exercises. We expanded the number of participants to 40: 20 rehabilitated with the proprioceptive elements and 20 with a conservative rehabilitation program, and prolonged the rehabilitation protocols to 6 months. Despite the increased number of participants and prolongation of the rehabilitation, the hypothesis that proprioceptive training would improve functional activity more than strengthening exercises was not supported.

Recent studies by Kaya et al. on patients after the ACL reconstruction using tibialis anterior allografts, found that the proprioceptive training could significantly improve quadriceps and hamstring muscle strength after 6 months of training [15]. Additionally, Risberg et al., who studied patients after arthroscopic reconstruction of the ACL using an autogenous bone-patellar tendon-bone, found that the neuromuscular training resulted in improvement in the functional tests [16]. A specific neurocognitive and perceptive rehabilitation treatment was more efficient that the common physical therapy in the Cappellino et al. study on patients after ACL reconstruction with patellar tendon [17]. These results are in clear contradiction to our conclusions. Moreover, all of these researchers found no statistically significant differences between studied groups in the subjective assessments. The strength of our study is that we have found a significant statistical relationship between the results of the subjective evaluation of the knee function and the values of the difference in the tibia shift in relation to the thigh, which confirms the important role of mechanical stabilization of the joint for its stability. This argues in favor of treating anterior instability with ACL reconstruction.

In the presented work, only one patient received a score below good in the Lysholm scale, and eight subjects in IKDC scale. All of these patients with low self-esteem of the function of the operated limb were the subjects from the group rehabilitated without the elements of proprioceptive training. They also had lower values of strength and endurance parameters of the knee extensors. These results are similar to those presented in the literature on the assessment of proprioception and knee function [10]. In our opinion, a low self-esteem was associated with worse knee function (lower values of the strength and the endurance parameters in the isokinetic study). This may be related to the ineffective work of the neuromuscular control system.

We found that the knee joint static stability decreased in three subjects. However, all subjects with such incomplete mechanical stabilization obtained results above "good" in the subjective evaluation scales. Such data indicate that the restoration of the mechanical stability is important, but not necessary for the recovery of good subjective function of the knee joint. This is also confirmed by the observation of the patients with diagnosed ACL damage, but without subjective symptoms of instability of the knee joint, so-called "copers" [18].

The isokinetic assessments between the studied groups showed no statistically significant differences for the operated limb. However, the healthy limbs of the patients from the group rehabilitated with the proprioceptive exercises were clearly stronger, both in the area of extensors and flexors. A muscle strength deficit of the operated limb in relation to the healthy one exceeding 20% was observed in 75% of the studied cases from this group. Additionally, the proprioceptive group from Liu-Ambrose's study demonstrated a greater percent change in isokinetic torques [13]. Moreover, de Jong et al. showed that the quadriceps strength deficit related to ACL damage usually remained at the level of 20% of the deficit after the ligament reconstruction [19].

To our knowledge, this is the first randomized controlled trial examining differences between rehabilitation programs including such a broad spectrum of techniques. In previous studies, studies used 1–5 assessing techniques and a portion of them focused only on the proprioception test. In this report we have combined patient-reported scores with the isokinetic strength parameters and mechanical stability examination, as well as visual proprioceptive control investigation. The strength of this study, as compared with the previous ones, is the length of the rehabilitation program, the number of the subjects included in each group and the thorough description of both rehabilitation programs.

Some limitations of the study also need to be addressed. The group of subjects was heterogeneous in terms of internal knee joint damage. The study included patients with an isolated ACL injury (19 in total: 11 in the group A and 8 in the group B) and with other internal knee injuries (21 subjects). However, this lack of homogeneity did not affect the assessment of the knee function one year after reconstruction of the ACL, since no statistically significant differences between the results of studies in patients with isolation of the injury and patients with accompanying internal injuries of the knee were noticed. The conclusion that the accompanying ACL damage, such as damage to the meniscus or III-degree cartilage, does not affect the results of subjective scales, proprioception scores [20] and the results of isokinetic evaluation [21] was drawn by other researchers as well.

## 5. Conclusions

The function of the knee one year after the ACL reconstruction was good or very good in the opinion of 80% of patients, no matter which type of rehabilitation program was included. Reconstruction of the ACL restored mechanical stability of the knee in 92% of all patients. After the ACL reconstruction, 70% all of patients presented abnormal or irregular visual proprioceptive control, despite rehabilitation programs. Turning off visual control definitely worsened the balance.

**Author Contributions:** Conceptualization, K.C.-G. and T.P.; methodology, K.C.-G.; formal analysis, P.B., K.C.-G., Ł.S. and K.B.-Ż.; investigation, K.C.-G.; data curation, K.B.-Ż., P.B.; writing—original draft preparation, K.B.-Ż.; writing—review and editing, K.B.-Ż. and P.B.; supervision, T.P. All authors have read and agreed to the published version of the manuscript.

**Funding:** This research received no external funding.

**Institutional Review Board Statement:** The study was performed with the approval of the local research ethics committee (Bioethics Committee at the Karol Marcinkowski Poznan University of Medical Sciences), in accordance with the Declaration of Helsinki, and all participants provided their written informed consent of participation in this study.

**Informed Consent Statement:** Informed consent was obtained from all subjects involved in the study.

**Data Availability Statement:** Data available on request from the corresponding author.

**Conflicts of Interest:** The authors declare no conflict of interest.

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
