# Peer review of "Similar Outcomes and Satisfaction of the Proprioceptive versus Standard Training on the Knee Function and Proprioception, Following the Anterior Cruciate Ligament Reconstruction"

_applsci, doi:10.3390/app11083494_

Round 1
Reviewer 1 Report
This research finding demonstrates that there is no advantage to function in proprioceptive rehabilitation exercises comparing to a traditional strengthening program. The statistical method appears to be sound and the conclusions are reasonable.
Major comments:
- Adding scatter plots to depict the correlation between subjective and objective results.
- Current objective tests do not reflect subjective results. What does this research really elucidate except that there is no advantage of proprioceptive rehabilitation excercise?
Author Response
Reviewer 1 - Comments and Suggestions for Authors
This research finding demonstrates that there is no advantage to function in proprioceptive rehabilitation exercises comparing to a traditional strengthening program. The statistical method appears to be sound and the conclusions are reasonable.
Our response: We appreciate this comment.
Adding scatter plots to depict the correlation between subjective and objective results.
Our response: The correlation between subjective and objective measurements is presented in a separate paragraph and statistically important results are presented in Table 6. Due to the correlation found only between 4 parameters and the large data of parameters tested, we decided not to include the scatter plots, since they will not add scientifically to the manuscript. We believe that presenting significantly important correlations will emphasize their importance.
Current objective tests do not reflect subjective results. What does this research really elucidate except that there is no advantage of proprioceptive rehabilitation excercise?
Our response: Current objective tests do not fully reflect subjective everyday situations. This study elucidates the importance and necessity of questionnaires in the evaluation of knee function. Therefore, we believe it should be advised to evaluate both objective and subjective tests in patients suffering from knee dysfunctions on a regular basis.
Reviewer 2 Report
In this manuscript, the authors report no advantage to function in doing proprioceptive rehabilitation exercise program following anterior cruciate ligament (ACL) reconstruction compared to traditional strengthening program. The authors have provided adequate details of the rehabilitation program and documented adequate knee scores.
It is moderately written and can benefit from professional English-language editing, as at times grammatical errors hinders its’ flow.
The methodology provides no randomization methods. Certain details of groups and demographics need to be added (for eg. BMI, level of activity prior to injury etc.) and moved to results section. I would like to see the authors comment on dominant lower limb rather than left or right. Also, can the authors clarify if during the procedure whether the ACL stump was preserved and if any platelet rich plasma injections were done at base of ACL.
Please provide p-value instead of n.s. The authors indicate that data is not in normal distributions but they use mean in tables. Maybe they can use median instead.
In the discussion, the authors need to clarify what was causing low self-esteem in subjects with scores below good for Lysholm and IKDC scales. Also, authors need to clarify what "knee joint had decreased.." means.
Authors need to mention whether their citations used hamstring vs bone-tendon-bone as their ACL reconstruction graft choice since this will affect quadriceps function for eg: reference 19 and not be a good comparison since the present manuscript uses hamstring graft only.
Overall, the manuscript in its present form has several limitations and I hope the authors will work on the aforementioned suggestions.
Author Response
Reviewer 2 - Comments and Suggestions for Authors
In this manuscript, the authors report no advantage to function in doing proprioceptive rehabilitation exercise program following anterior cruciate ligament (ACL) reconstruction compared to traditional strengthening program. The authors have provided adequate details of the rehabilitation program and documented adequate knee scores.
Our response: We appreciate this comment.
It is moderately written and can benefit from professional English-language editing, as at times grammatical errors hinders its’ flow.
Our response: The manuscript was edited by Native Speaker.
The methodology provides no randomization methods. Certain details of groups and demographics need to be added (for eg. BMI, level of activity prior to injury etc.) and moved to results section.
Our response: We have re-analyzed the presented data and correlated the results with the body mass index and the age of the patients. We have found that the self-assessment of the knee joint function depends on the age and BMI of the participants and does not depend on the rehabilitation program. This result was included in the Results section, as suggested (page 4, lines: 140-147).
I would like to see the authors comment on dominant lower limb rather than left or right.
Our response: We have changed the description of the limbs, as suggested (page 2, lines 67 and 73).
Also, can the authors clarify if during the procedure whether the ACL stump was preserved and if any platelet rich plasma injections were done at base of ACL.
Our response: The ACL stumps were removed and we didn't use platelet rich plasma injections. We have introduced this clarification on page 3 (lines 85-86).
Please provide p-value instead of n.s.
Our response: We have included the p value instead of n.s. in Tables 2, 3, 4 and 5, as suggested.
The authors indicate that data is not in normal distributions but they use mean in tables. Maybe they can use median instead.
Our response: This is correct, we have presented median value and not the mean value as depicted, as stated in the Methods section. We have corrected incorrect labeling of the tables 2, 3, 4 and 5.
In the discussion, the authors need to clarify what was causing low self-esteem in subjects with scores below good for Lysholm and IKDC scales.
Our response: In our opinion, a low self-esteem was associated with the worse knee function (lower values of the strength and the endurance parameters in the isokinetic study). This may be related to the ineffective work of the neuromuscular control system. We have introduced this explanation in the Discussion section, as suggested (page 7, lines: 230-233).
Also, authors need to clarify what "knee joint had decreased.." means.
Our response: We are very sorry for this unfortunate phrase. We meant “knee joint static stability had decreased”. We have corrected this on page 7, line 234.
Authors need to mention whether their citations used hamstring vs bone-tendon-bone as their ACL reconstruction graft choice since this will affect quadriceps function for eg: reference 19 and not be a good comparison since the present manuscript uses hamstring graft only.
Our response: We have added these missing information in the Discussion section – page 7, lines: 204-205, 214-215, 217-218 and 221-222.
Overall, the manuscript in its present form has several limitations and I hope the authors will work on the aforementioned suggestions.
Our response: We have changed the manuscript following the Reviewers’ suggestions and we hope you will find it suitable for publication in the present form.
Round 2
Reviewer 2 Report
The authors have done a great job in addressing my concerns. However, there are a certain things that need to be clarified before publication.
Introduction
Line 26 - anterior cruciate ligament (ACL)
Line 27 - can consider rephrasing - a recent meta-analysis by ___ et al revealed that ...
Line 29 - can rephrase - Furthermore,..... have also been reported.
Line 31 - can rephrase - ' the speed and safety with which patients return to pre-injury level of function post-ACL reconstruction
Line 33 - evidence suggests that an accelerated rehab protocol ...., however ... no consensus on such program...
Line 41 - can consider rephrasing: recent trends in acl rehab are consistently using proprioceptive training with mixed results. therefore further work in required to determine.....
Line 42, 44, 47 - use ACL - please use abbreviations consistently throughout the manuscript
Methods
Line 51: consider rephrasing - patient >6m post injury who underwent acl recon using a hamstring graft (...) at university.... by one experienced paediatric orthopaedic surgeon from year .... were included in the study. patients with coexisting damage....trauma were excluded.
patients were randomized to two groups - group a which consisted of ..... rehab protocol and group b ....
randomization was done using ....
Line 83: no need to write PRP, that was for my clarification
Line 96: shortly not needed, can start with in the first stage...
line 97: were introduced on post op day 2
this section on rehab protocol needs to be made clear with proper stage labels. I loose the flow when you mention in next stage... it should be in the first stage ..., in the second stage... furthermore addition to the stages these....
Results
make a demographics section and a table
Forty patients were included in the study. ... age, .... were male, ... was bmi, ... pre-injury scores (if any), .... dominant leg....
then go to subjective and objective eval scores
Discussion
line 194: the study saw no difference between .....
line 206 - strengthening 'alone'
line 252 - please provide randomization technique in methods.
Author Response
Reviewer - Comments and Suggestions for Authors
Line 26 - anterior cruciate ligament (ACL)
Our response: We have rephrased this sentence.
Line 27 - can consider rephrasing - a recent meta-analysis by ___ et al revealed that ...
Our response: We have rephrased this sentence.
Line 29 - can rephrase - Furthermore,..... have also been reported.
Our response: We have rephrased this sentence.
Line 31 - can rephrase - ' the speed and safety with which patients return to pre-injury level of function post-ACL reconstruction
Our response: We have rephrased this sentence.
Line 33 - evidence suggests that an accelerated rehab protocol ...., however ... no consensus on such program...
Our response: We have rephrased this sentence.
Line 41 - can consider rephrasing: recent trends in acl rehab are consistently using proprioceptive training with mixed results. therefore further work in required to determine.....
Our response: We have rephrased this sentence.
Line 42, 44, 47 - use ACL - please use abbreviations consistently throughout the manuscript
Our response: We have rephrased this sentence.
Methods
Line 51: consider rephrasing - patient >6m post injury who underwent acl recon using a hamstring graft (...) at university.... by one experienced paediatric orthopaedic surgeon from year .... were included in the study. patients with coexisting damage....trauma were excluded.
patients were randomized to two groups - group a which consisted of ..... rehab protocol and group b ....
randomization was done using ....
Our response: We have rephrased this section.
Line 83: no need to write PRP, that was for my clarification
Our response: We have deleted this sentence.
Line 96: shortly not needed, can start with in the first stage...
Our response: We have rephrased this sentence.
Line 97: were introduced on post op day 2
Our response: We have rephrased this sentence.
this section on rehab protocol needs to be made clear with proper stage labels. I loose the flow when you mention in next stage... it should be in the first stage ..., in the second stage... furthermore addition to the stages these...
Our response: We have rephrased this section.
Results
make a demographics section and a table
Forty patients were included in the study. ... age, .... were male, ... was bmi, ... pre-injury scores (if any), .... dominant leg....
then go to subjective and objective eval scores
Our response: We have rephrased this section. We have not analyzed pre-injury scores nor leg dominancy.
Discussion
line 194: the study saw no difference between .....
Our response: We have not rephrased – please explain more precisely where to apply corrections.
line 206 - strengthening 'alone'
Our response: We have not rephrased – please explain more precisely where to apply corrections.
line 252 - please provide randomization technique in methods.
Our response: We have provided it in the methods.